# “In the Wild” Video Content as a Special Case of User Generated Content and a System for Its Recognition

**DOI:** 10.3390/s23041769

**Published:** 2023-02-04

**Authors:** Mikołaj Leszczuk, Marek Kobosko, Jakub Nawała, Filip Korus, Michał Grega

**Affiliations:** 1AGH University of Science and Technology, 30-059 Kraków, Poland; 2Department of Electrical Electronic Engineering, University of Bristol, Bristol BS8 1QU, UK

**Keywords:** Quality of Experience (QoE), Quality of Service (QoS), metrics, evaluation, performance, Computer Vision (CV), Video Quality Indicators (VQI), Key Performance Indicators (KPI), User-Generated Content (UGC), in the wild content

## Abstract

In the five years between 2017 and 2022, IP video traffic tripled, according to Cisco. User-Generated Content (UGC) is mainly responsible for user-generated IP video traffic. The development of widely accessible knowledge and affordable equipment makes it possible to produce UGCs of quality that is practically indistinguishable from professional content, although at the beginning of UGC creation, this content was frequently characterized by amateur acquisition conditions and unprofessional processing. In this research, we focus only on UGC content, whose quality is obviously different from that of professional content. For the purpose of this paper, we refer to “in the wild” as a closely related idea to the general idea of UGC, which is its particular case. Studies on UGC recognition are scarce. According to research in the literature, there are currently no real operational algorithms that distinguish UGC content from other content. In this study, we demonstrate that the XGBoost machine learning algorithm (Extreme Gradient Boosting) can be used to develop a novel objective “in the wild” video content recognition model. The final model is trained and tested using video sequence databases with professional content and “in the wild” content. We have achieved a 0.916 accuracy value for our model. Due to the comparatively high accuracy of the model operation, a free version of its implementation is made accessible to the research community. It is provided via an easy-to-use Python package installable with Pip Installs Packages (pip).

## 1. Introduction

According to Cisco, IP traffic increased by a factor of three between 2017 and 2022. In 2022 IP video traffic represented 82% of all IP traffic. IP video traffic increased from 91.3 Exabytes in 2017 to 325.4 Exabytes in 2022 in terms of monthly traffic. Furthermore, Cisco estimates that in 2022 the average Internet user produced 84.6 gigabytes of bandwidth per month. As a result, fixed Internet traffic from the client per user per month increases by 194% from 28.8 gigabytes in 2017. Finally, we notice an improvement in video quality: By 2023 66% connected flat-screen TVs are 4K according to [1]. User-Generated Content (UGC) is mainly responsible for IP video traffic created by the users. First, we provide a clearer understanding of what UGC means. According to definitions in the literature, UGC, also known as User-Created Content (UCC), is any type of content that users submit on on-line platforms (such as social media platforms forums and Wikis). However, we regard UGC as just the content that users develop and upload themselves from the perspective of content quality. Furthermore, there are alternative meanings that are totally unrelated to the context of video; UGC is a product that users produce to transmit information about items online or the businesses that promote them according to [2,3]. UGC opposes Professional-Generated Content (PGC) and Marketer-Generated Content (MGC) [4].

### 1.1. Research Scope

A Quality of Experience (QoE) assessment may occasionally be necessary for UGC, just as is necessary for any other type of content. Evaluation of UGC quality is a hot research topic right now. The query about UGC video quality produces more than 76,000 results in Google Scholar. As a result, we may assert that there is a substantial body of studies on the calibre of UGC. After all, it is not surprising that the algorithms for UGC quality evaluation are obviously different from general quality evaluation due to the peculiarity of the content. However, the question arises before we even begin to evaluate the UGC’s quality: How can we tell if the content is truly user-generated? There are not many studies on UGC recognition. To our knowledge, there are no true operational algorithms that separate UGC content from other content, according to studies in the literature. The problem is made more difficult by the fact that in the modern world, the status of a piece of content as a UGC is no longer dictated by the circumstances surrounding its acquisition or the methods used to handle it. The development of widely accessible knowledge and affordable equipment makes it possible to create quality UGCs that are almost indistinguishable from professional content, although at first this content has been frequently characterized by amateur acquisition conditions and unprofessional processing [5]. We are less interested in all UGC content because some of it is difficult to differentiate from professional work than we are in UGC which has a distinct difference in quality from professional content. For the purposes of this paper, we refer to “in the wild” as a closely related idea to the general idea of UGC, which is its particular case.

For example, in the case of news content, it is crucial to distinguish between “in the wild” and professional content. Although the news is mainly professional content, it can also feature “in the wild” content and may also vary in “saturation levels”. Figure 1 which provides six examples of the integration of professional content with “in the wild” content in the news serves as an illustration of these (Video source:https://youtu.be/8GKKdnjoeH0, https://youtu.be/psKb_bSFUsU and https://youtu.be/lVuk2KXBlL8, accessed on 31 December 2022). As one can see, it is possible to have shots: only with professional content (Figure 1a) with professional content but with a relatively small display presenting the “in the wild” content (Figure 1b) with professional content but with a relatively large display presenting the “in the wild” content (Figure 1c) with “in the wild” digitally mixed with a large area of professional content (Figure 1d) with “in the wild” content digitally mixed with a small area of professional content (Figure 1e) and finally exclusively with the “in the wild” content (Figure 1f).

In this research paper, we demonstrate that the novel idea of an objective “in the wild” video content identification model can be introduced. The accuracy of our model reaches 0.916. This model is trained and tested using video sequence databases that comprise both professional and “in the wild” content. As contributions of this paper we:Present motivation on what the problem is and why it is important (addressed in Section 1);Show our objectives (addressed in Section 1.1);Reveal gaps in the literature (addressed in Section 1.2);Propose to fill some of these gaps (addressed in Section 1.3);Present overview on our overall approach (addressed in Section 2);Show results contributing to the literature and to the video analysis research area in general (addressed in Section 3);Discuss novelties of the paper (addressed in Section 4).

### 1.2. Related Work for Multimedia Quality Assessment

To begin, we must first clarify what we mean by *“in the wild” content*.

The main characteristic of “in the wild” content is that it is not carefully curated. The movements of cameras in “in the wild” content are often erratic. This frequently happens when content is captured on a portable device (e.g., a smartphone).

Furthermore, it leads us to limit the use of the definition to the setting of the QoE or the Multimedia Quality Assessment (MQA) influenced by contextual and human systemic factors as shown in Figure 2 by Karadimce and Davcev [7].

To start, we give the reader a definition of the term “in the wild” from the Cambridge Advanced Learner Dictionary & Thesaurus. According to the dictionary, the word means “under natural conditions independent of humans”. The reference to natural circumstances mirrors how the word “in the wild” is used in the MQA literature although the second part of the definition does not apply to the scenario being explored here. Observe that the naturally occurring content contrasts with artificially distorted content [8].

As the name implies, the undistorted content is first recorded before being artificially warped. The chosen distortions are then *artificially* introduced. The experimenters typically choose the distortions manually with the goal of imitating impairments that might occur in the real world. The content of “in the wild” has flaws that are already evident in the real world. There is a growing trend in MQA research to use content found “in the wild” rather than artificially altered content because doing so always increases the ecological validity of the tests. The authors of [8] also mention that there is no access to the unaltered version of the content because it is already compromised “in the wild”. As a result, it is not possible to use the well-known and respected Full Reference (FR) techniques in MQA.

The content of “in the wild” characteristics corresponds to the content of the real world according to the authors of the cited paper [9] (in contrast to the synthetically distorted content). They also use the terms “in the wild video” and “UGC video” interchangeably, which is significant. The term “real-world in the wild UGC video data” is also used by Ying et al. to describe a category of video data that their data set reflects.

Taking this into account, we can draw the conclusion that [9]’s authors do not distinguish between UGC videos and “in the wild” videos. Instead, the authors contend that UGC videos serve as examples of real-world content categories. This is backed among other things by the fact that UGC videos suffer from a variety of intricate distortions (e.g., unsteady hands of content creators processing and editing artefacts or imperfect camera devices).

Tu et al. [10] do not give a clear description of what constitutes “in the wild” content. Instead, they mix the idea with UGC videos [9]. They specifically specify “in the wild UGC databases”.

However, the authors of [10] contrast natural UGC databases with artificial single-diffusion databases (the latter representing a legacy approach and being less representative of real-world conditions). According to them, compression artifacts are not often the main elements affecting video quality in real-world UGC databases. This finding further distinguishes “in the wild” UGC databases from older databases, which often emphasize compression artifacts (sometimes complementing them with transmission artifacts).

Yi et al. [11] combine the ideas of UGC and “in the wild” videos following the precedents of [9,10]. Like [9,10] they draw attention to the fact that UGC recordings exhibit intricate distortions of the real world. They provide new information such as the fact that nonuniform spatial and temporal aberrations are present in “in the wild” UGC videos. Compared to old artificially distorted video databases here distortions are applied consistently over an image or a video frame even if more than one distortion is used.

In addition to the UGC videos, we also consider the so-called eyewitness footage in our work. However, there is at least one element that eyewitness footage and UGC videos have in common. In other words, they are taken in a natural setting. Therefore, the videos that we consider in this work are called *in the wild video*. This is significant because it differs from other MQA works that frequently mix the terms “in the wild” and “UGC” and refer to the data they use as “in the wild UGC videos”.

However, it should be noted that a review of the relevant literature suggests that the issue of identifying UGC content has already been discussed but not in relation to content that is “in the wild”. Additionally, not all references to multimedia content are necessarily references to UGC content. The paper by Egger and Schoder [12] can be given as an illustration; it discusses the categorization of the text. However, even the approaches for categorizing multimedia content that is revealed during the literature research are not always based on video data. In contrast to the model offered in [13] which uses video while classifying it, the model described in [14] for instance uses audio attributes.

### 1.3. “In the Wild” Definition

Now we provide our definition for the *“in the wild” video* content category. This definition serves to draw attention to content that we believe is pertinent to the work at hand. The easiest way to define “in the wild” videos is to say that they were recorded and edited in a natural setting. Videos shot “in the wild” specifically meet the needs listed below.
They are recorded without the use of professional equipment. In particular without using a professional camera to produce videos.They are not recorded in a studio. In other words, they are captured in an environment without an intentional lighting setup.They are shot without a gimbal or similar image stabilization equipment. In short, these are handheld videos.They are not subjected to significant post-processing aimed at intentionally improving their quality. In other words, they are not post-processed at the film production level.

The remainder of this paper is organized as follows. The creation of a corpus of databases the original video sequences made up of already existing footage and the creation of the model are described in Section 2. Section 3 contains a report of the results. The findings of this paper are discussed in Section 4.

## 2. Materials and Methods

This section describes database construction (Section 2.1) and their detailed description (Section 2.2). Then the section describes the video indicators we use for modeling (Section 2.3) and the model itself (Section 2.4).

### 2.1. Database Construction

Our model (as described below) is based on supervised learning, so the first step of the investigation is to prepare a database that is used to train and test the algorithm. For this purpose, the individual video sequences are divided into shots. Then from each fragment, the average of the video indicators (described below) is determined, and the shots are manually divided into individual categories.

The assignment to a given category is based on the criteria given above.
Professional video sequences are those recorded in a film studio or outside using professional equipment with image stabilization. These video sequences are also often post-processed.“In the wild” video sequences can best be described as being recorded and processed under natural conditions.

The database obtained has 12,067 results, of which 3535 belong to the amateur category (“in the wild”). The database is made available to fellow researchers to download in the section “Appendix A”.

### 2.2. Database Description

The database is created based on content from news channels and amateur (“in the wild”) video sequences posted on YouTube.

Each video is cut into shots. First, the frame rate of the video is calculated using opencv module. Then using scenedetect module the list of scenes in the video is calculated. Then the moviepy module helps to separate the shot from the video based on the scene list.

For each frame in the shot, indicators are calculated. Then the mean of the gathered indicator values for the shot is calculated. Then every shot is assigned to the right category by hand.

The channels from which the source films come are diverse. The database consists of materials from the Euronews France24 RT Saudi TV and BBC TV stations. Both professional and amateur shots (“in the wild”) were extracted from these sources. Due to the small number of amateur shots (“in the wild”) in such content, the database is expanded with content from the Web.

To keep the database as universal as possible, content of various resolutions is used. The examples in the database are stored in their original resolutions. Table 1 shows the video resolutions and the distribution of the samples.

### 2.3. Video Indicators

The configuration of the video indicators is described in this section. It gives a general overview and describes the various video indicators that are used.

The AGH Video Quality (VQ) team previously created a software program to measure quality indicators, which operates in a challenging NR model. The idea of visual quality monitoring based on key performance indicators (KPI) that was previously created is now realized in this software package according to [15]. The concept presented here is known as artifacts (or KPIs), which are broken down into four groups based on where they came from: a category for capturing processing, transmitting, and displaying. In addition to helping to set the algorithm, extend the monitoring intervals, and ensure improved QoE prediction, the program can isolate and improve event investigation [16].

We employ a total of 10 video indications. One can download them by clicking the link in the section titled “Appendix A”. All of the video indicators used to train the system are listed in Table 2. It also references papers that are relevant. Examples of “in the wild” content and professional content with respect to the features listed in Table 2 are given in Appendix B.

Given how close the frames are to each other, they have comparable values for the video indicators. As a result, the experiment uses averaged video frames. It uses several video indications and generates a vector of outcomes (one for each video indicator). The results are then compared with the ground truth data (“in the wild” as opposed to professional content). They make up the input data for modelling when combined.

### 2.4. Model

This Subsection describes algorithm (Section 2.4.1) modelling and parameters used (Section 2.4.2) determination of hyperparameters (Section 2.4.3) as well as model training (Section 2.4.4).

#### 2.4.1. Algorithm

First, we provide some information about the algorithm itself. XGBoost (Extreme Gradient Boosting) is a machine learning algorithm that is used to perform both regression and classification. It is an implementation of the gradient boost technique in R and Python. XGBoost stands out for its high performance and good results, due to the use of several different techniques, such as subsetting and regularization.

In Python, the XGBoost classification is performed using the XGBClassifier class. When training the XGBClassifier model, the XGBoost algorithm performs an iterative performance boost by applying the gradient boost technique to successive subsets of the training data. The final result is a set of models that can be used for classification.

To further improve the performance of the XGBClassifier model, additional hyperparameters can be configured, such as the maximum number of trees in the model or the regularization factor. This allows the model to be adapted to specific needs and reduces the risk of overfitting.

In summary, the XGBoost algorithm and its XGBClassifier class are effective tools for gradient enhancement classification. Using various techniques, such as subsetting and regularization, XGBoost provides high performance and good results.

#### 2.4.2. Description of Modeling and Parameters Used

Parameter modelling in XGBClassifier is performed using a parameter grid (param_grid). The parameter grid allows one to specify different values for each model hyperparameter and experiment to find the best settings for the problem. XGBoost [21] model was tuned based on the following parameters:“max_depth”“scale_pos_weight”“learning_rate”“reg_lambda”“colsample_bytree”“gamma”

#### 2.4.3. Determination of Hyperparameters

To determine the optimal values of the parameters GridSearchCV from Scikit-learn is used. The scikit-learn library’s GridSearchCV function does an exhaustive search for an estimator over a defined-parameter grid. The function trains an estimator for each combination of hyperparameters on the grid and evaluates it using cross-validation. Then it returns the best parameters and the best score from the given grid.

The argument scoring used in GridSearchCV specifies the metric that will be used to evaluate the models during the grid search. In this case, the metric is the area under the receiver operating characteristic curve (AUC). The model with the highest AUC score will be considered the best model. The number of cross-validation folds to use when evaluating each model is defined by the argument cv and was set to 3.

When the function is run multiple times and the parameters it returns are adjusted, the best parameters are found for the given set of hyperparameters. In this case, the tuned hyperparameters present themselves as follows:“max_depth”: 10“scale_pos_weight”: 0.6“learning_rate”: 0.12“reg_lambda”: 0.7“colsample_bytree”: 0.7“gamma”: 0.3

Subsequently, the parameters determined are applied to the model and cross-validation is performed on it using the K-Fold module from Scikit-learn. K-Fold cross-validation is used to assess the generalization performance of a model. The database is divided into 10 K folds for cross-validation using the KFold class. To perform cross-validation on a given estimator, the cross_val_score function from Scikit-learn is used.

The other approaches available for cross-validation are leave-one-out cross-validation (LOO-CV) and leave-one-user-out cross-validation (LOUO-CV). These approaches were clearly described in [22]. We chose the k-fold method instead of the other two on the basis of the size of the database. Both calculation times for the leave-one-out approaches took too long to be sustainable in our tests.

The cross-validation accuracy result is 91.70%. For comparison, not using the optimal parameters and using their default values results in an accuracy of 89.77%.

#### 2.4.4. Model Training

The collected data are divided into the 80/20 proportion in a training and validation set, and then a classification algorithm based on it is trained.

XGBClassifier with a defined binary purpose is used as the base algorithm because the target labels are 0 or 1. This corresponds to the logistic regression objective, which is to minimize the binary cross-entropy loss.

We use the evaluation set consisting of validation set data to define when it stops improving. To track model performance during training, log loss metric is used. This helps to identify when the model is over-fitting or under-fitting and to define the value of early_stopping_rounds parameter. It is used to prevent overtraining of the algorithm, as the model will not be trained on data that do not improve its generalization performance. In this case, the early_stopping_rounds parameter of 10 is used.

A list of random variables is generated to be used as random_state when splitting into training and validation sets. Then training and prediction are performed 20 times in the loop. This is done to cross-validate the model. We chose to use the loop of 20 to reduce the impact of randomness. By doing so, we lowered the standard deviation of the results and stayed within a reasonable calculation time.

Each time, the accuracy precision and recall of the model’s predictions are measured and appended to the list. From this average accuracy value, the highest accuracy value, the lowest accuracy value, standard deviation, precision, recall, and the F1 score are calculated. After evaluation, the model that achieved the best results is exported and tested on different data sets.

## 3. Results

The accuracy results achieved for tuned modeling, as well as the comparison with other algorithms, are presented in Table 3.

Further evaluation of the results has shown that model prediction on “in the wild” samples characterizes with lower accuracy than professionally generated samples. The results of the average classification report are presented in Table 4. The test set contained 2414 samples. To better illustrate the classification, we used a confusion matrix for the presentation of the results; it was presented in Figure 3 where label 0 represents “in the wild” content and the label 1 professionally generated content.

Furthermore, we created a table of feature importance to present which parameters were significant for the calculations. The table was created using the feature_importances_ function available from xgboost and is presented in Figure 4. Insight on the features that are relevant to the model allowed us to further investigate the misclassified samples and their parameters.

From the testing set, we separated “in the wild” samples from professionally generated ones. Furthermore, we split the samples into correctly and wrongly classified, creating four sets of data. Then, for each set, we calculate the mean values of the video indicators in order based on the importance of the features. We also calculated the mean indicator values for “in the wild” and professionally generated samples for all samples in the dataset. This allowed us to compare the mean values of the data set with the mean values of the each of four sets separated from the testing data set. To present the change, we calculate the percentage deviation of each indicator in the sets of averages, and the results are presented in the following Table 5. The indicators are displayed in order according to Figure 4. The equation used for the deviation calculation is as follows:(1)$$\left|\frac{100(B-A)}{A}\right|\%$$
where *A* is the mean value of the indicator of all samples in the dataset, and *B* is the mean value of the indicator for each test set.

Minimal indicator deviation is the desirable state because then the model has a clear path on how to classify the indicator. As we can see, some of the indicators have high percentage deviation and high importance levels at the same time. This method helped us determine which indicators have the greatest impact on misclassification and to better understand the impact of each indicator.

By analyzing the percentage deviation of future importance and the results based on the category, we conclude that
“In the wild” samples are more difficult to classify.The indicators responsible for the errors are as follows. Temporal Activity Block Loss Spatial Activity Slicing Blur and Noise.The two most relevant indicators that cause errors were Temporal Activity and Block Loss.

We realize that some of the samples deviate very much from the average, and thus cause errors in classification. The database is highly universal, and it does not contain perfect samples but rather a wide spectrum of data.

In some cases, the resolution and quality of the “in the wild” content is very high, which blurs the boundary between professionally generated content. The same goes the other way. This phenomenon confuses the model, and by doing so reduces its accuracy.

To evaluate our machine learning model with deep neural networks proposed in [23,24,25] we decided to implement convolutional neural networks (CNN) and long-short-term memory layer (LSTM) and for our problem. We chose to use 1 dimension CNN based on the characteristics of the dataset. During training logarithmic loss function (binary_crossentropy) was used, as well as the Adam optimization algorithm for gradient descent.

To experiment with CNN, two models were created, one of which implemented the LSTM layer. Both models use multiple layers consisting of two 1D convolutional layers, two max-pooling layers, two dropout layers, one flattened layer, one dense layer and in CNN-LSTM’s case, one LSTM layer. Both models were trained and evaluated on the same dataset as XGBoost using 80/20 train, test split. In each compilation, 50 epochs were used. Training and testing were performed 20 times, the same as for other algorithms. The mean results of the prediction on the test set are presented in Table 6.

Proposed neural networks are superior in the field of precision, recall, and F1-score. However, XGBoost presents higher accuracy results than deep learning models.

The results were compared with [26]. The authors presented a different approach and identified the samples based on video and audio recognition, contrary to our method, which contains only video. The method was used for video summarization. Using a random forest classification they achieved  65.3% accuracy.

On the other hand, the authors of [23] focused on creating a new deep learning architecture called the recurrent convolutional neural network (RCNN). This architecture is capable of extracting both local and dense features from individual image frames, as well as learning the temporal features present between consecutive frames. This method achieved  86.3% accuracy.

One of the most popular approaches to image and video classification has been the use of CNN. CNN network presented in [24] achieves  96.07% accuracy. The authors focused on object classification from multiple shots of the same object. As a result, the collection of photos of the same object was categorized into one group and the use of multiple perspectives improved the accuracy.

CNNs were also used in [25] to summarize moments of iteration of sports videos. To achieve the goal, the authors also added the LSTM to the model. The paper focuses on transforming the video into a summary of action highlights. The results are accurate, but the evaluation is much more subjective and fully described in the paper.

The advantage of XGBoost in comparison to the methods described above is the high performance in training. It was one of the reasons why we decided to use this machine learning model instead of focusing on deep learning solutions. By comparing the results with other methods, we evaluate that usage of XGBoost and video indicators is not only superior performance-wise, but it presents high accuracy results.

The model trained this way is exported and tested on foreign data sets.

The first test set is a database that originates from [6]. It uses 3 publicly available “in the wild” video databases: (i) CVD-2014 [27] (ii) LIVE-Qualcomm [28] and (iii) KoNViD-1k [29]. The database of video sequences is supplemented with a “counterweight” in the form of professional quality video sequences. For this purpose, the database “NTIA simulated news” is used [30]. In the beginning, there is a high imbalance of samples in the class (“in-the-wild” content vs. professional content) of about 1:32, which is not recommended for modelling. Due to this, samples of the second class are randomly subsampled to obtain a more balanced ratio of 1:5. As a result, 408 samples are used for the database, which is divided into 68:304. The model achieves an accuracy of 94.5% in it.

The second test set contains 44 shots consisting of only professional samples. It was created based on the studio material from France 24 news, which was divided into shots. The resolution of the video is 480p. In this test, the model recognized each shot correctly.

To objectively assess the performance of the XGBoost model against competitive machine learning models, a series of measurements are performed. We experimented with different machine learning algorithms with similar execution times to find if XGBoost has the highest accuracy. Table 3 shows the accuracy results of each algorithm, while Table 7 shows the weighted average classification report results, all tested with a data split of 80/20. Similarly to training the XGBoost model, we chose to take measurements 20 times for each model.

Performance-wise, the execution of the algorithm can be divided into two parts (i) computation of the indicators and (ii) execution of the XGB Classifier. The performance of the indicator calculation is described in detail in [16]. Execution of all indicators on a single 1920 × 1080 frame in a single-threaded application on an Intel Core i7-3537U CPU takes 0.122 s. This time can be further reduced to 0.089 s by using multithreading. More efficient extraction algorithms are a work in progress, with a goal of reaching less than 0.033 s to process 30 FPS full HD video in real time.

Execution of the XGBClassifier on pre-computed vectors of indicators is extremely fast, several orders of magnitude faster than the calculation of the indicators, and we estimate that it reaches $9\times 10^{-5}$ s.

## 4. Discussion

In this paper, we show that it is possible to introduce the new concept of an objective “in the wild” video content recognition model. The value of the measured accuracy of the model achieved is 91.60%. We consider the results obtained to be very good. However, the model created obviously has limitations. First, it is limited to “in the wild“ content that meets the criteria described in Section 1.3. If the “in the wild” content is generated in a way that is indistinguishable from professional content, the model may fail to properly categorize the sample.

The functionality for automated detection of “in the wild” content may be important in many applications, such as content metadata enrichment Intellectual Property Rights (IPR) management, content quality management, and video summarization.

The development of “in the wild” content recognition will have a positive impact on the media industry by broadening the portfolio of methods that can be used for automated metadata extraction. In our opinion, automated “in the wild” content tagging has a broad range of applications.

For many content providers, it is important to have rich metadata for their content databases. Regardless of metadata format and organization, the richer the metadata, the easier it is to manage the content. Automated and robust detection of “in the wild” content allows one to enrich existing metadata with this new information without any significant effort. This may lead to the development of new applications as described below.

“In the wild” content identification allows one to identify and solve Intellectual Property Rights issues for content providers. If the media in the database is user-generated, it is crucial to identify and, if necessary, obtain the IPR before broadcasting or sharing. Otherwise, it is possible to violate the IPR of a third party, which may have legal consequences. When the “in the wild” content is automatically tagged, the probability of such events is significantly reduced.

Additionally, the identification of “in the wild” content allows easier content quality management. Most of the time, “in the wild” content has a lower subjective visual quality than professional content. Broadcasting such content may result in a lower overall quality of content offered. Having such content identified allows us to manage the situation when such content is broadcasted, e.g., using appropriate overlay graphics.

Finally, automated “in the wild” content identification allows one to create richer and more engaging summaries of video content. In our recent research [31] we have explored the possibility of using different video indicators to create a summary of news and reports. We have used indicators such as Spatial Activity (SA) and Temporal Activity (TA) (Furthermore, known as Spatial Intensity (SI) and Temporal Intensity (TI).) for this purpose. Now, we focus on incorporating the “in the wild” content identification. For this purpose, we divide the input video into individual shots using automated shot detection. Afterwards, we identify the “in the wild” content and professional content in the video using the solution described in this paper. We have tested several different methods to incorporate “in the wild” content into summaries. Although this is ongoing research, the preliminary results of user tests suggest that the use of “in the wild” indicators may improve the quality of the summary over those created with TA and SA only.

Last but not least, we develop and provide the Python package that is installable with Pip Installs Packages (pip) which provides easy-to-use functions to compute video indicators including the “in the wild” content indicator. More information about the package can be found in Appendix C.

By developing and delivering a Python package, we hope that as part of further work various groups of researchers will be able to create “in the wild” content quality assessment algorithms, which, as we mention, are, of course, different from general quality assessment algorithms due to the specificity of the content.

## Figures and Tables

**Figure 1 sensors-23-01769-f001:**
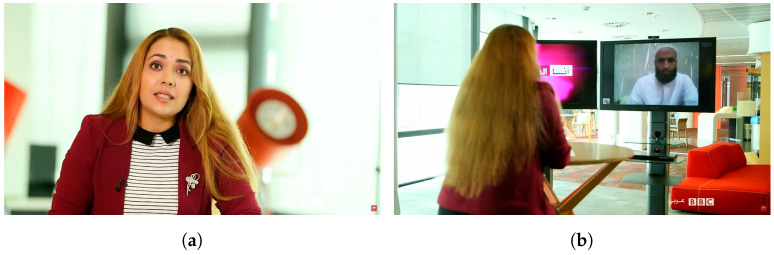
Professional content vs. “in the wild“ content (source: [6]). (**a**) Professional content with no “in the wild” content; (**b**) Professional content with “in the wild” content displayed in small area; (**c**) Professional content with “in the wild” content displayed in large area; (**d**) “in the wild” content with professional content mixed in large area; (**e**) “in the wild” content with professional content mixed in small area; (**f**) “in the wild” content with no professional content.

**Figure 2 sensors-23-01769-f002:**
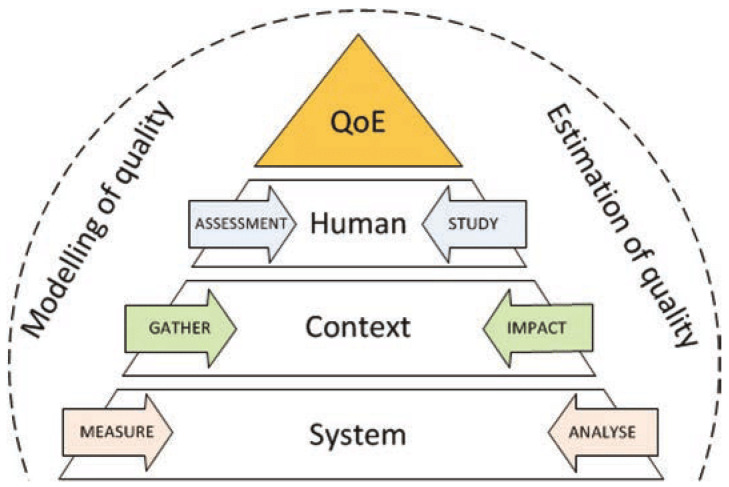
Multimedia quality evaluation methodology (source: [7]).

**Figure 3 sensors-23-01769-f003:**
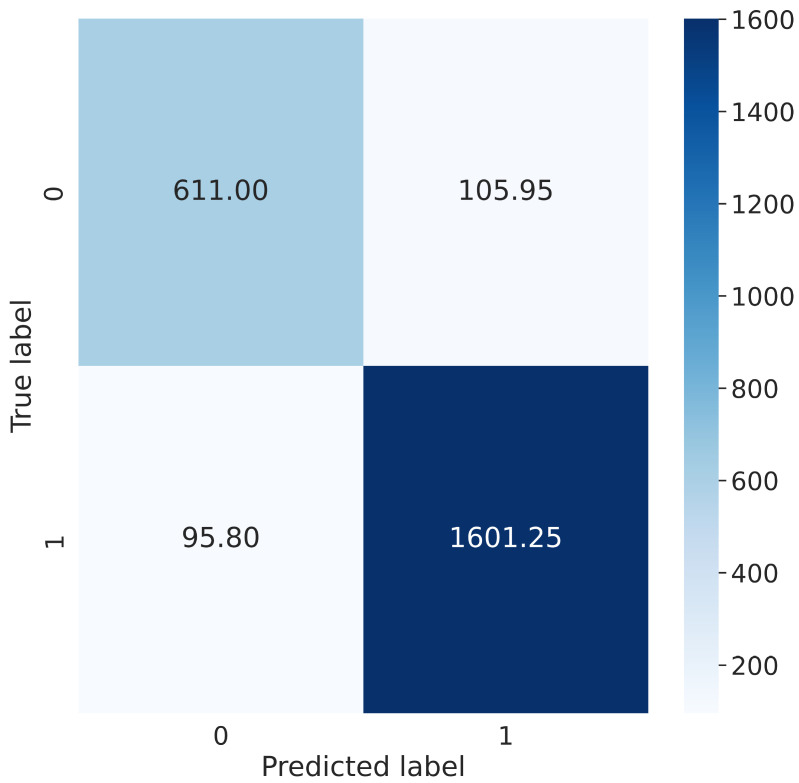
Confusion matrix of test set classification.

**Figure 4 sensors-23-01769-f004:**
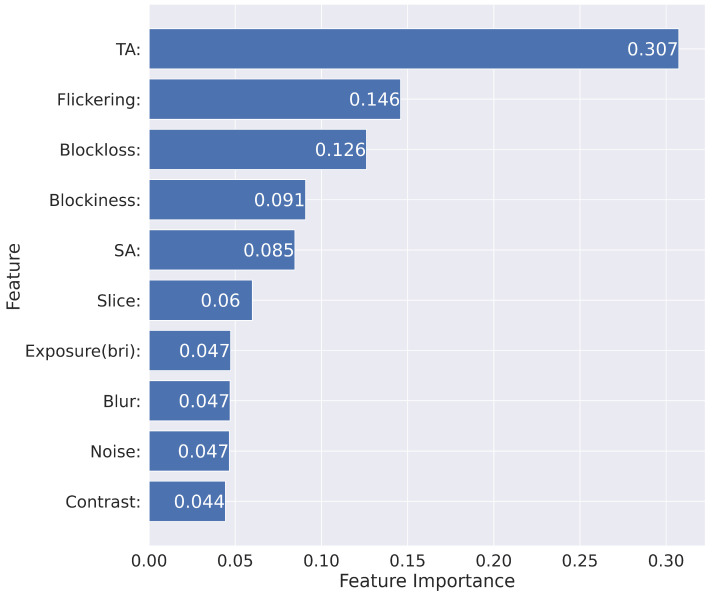
Importance of the XGBoost feature in sample classification.

**Table 1 sensors-23-01769-t001:** Sample format distribution.

Format	Number of Samples
144p	2512
360p	98
480p	310
720p	7893
1080p	1254

**Table 2 sensors-23-01769-t002:** The list of video indicators which are used in the experiment (based on: [6]).

#	Name	Description
1	Blockiness [17]	Block boundary artefacts, also known as checker-boarding, quilting, or macro-blocking
2	Spatial Activity (SA) [17]	The degree of detail in a video, such as the presence of sharp edges, minute details, and textures
3	Block Loss [15]	A vision artefact resulting in the loss of selected image blocks
4	Blur [17,18]	Loss of high spatial frequency image detail, typically at sharp edges, as a result
5	Temporal Activity (TA) [17]	The amount of temporal change in a video sequence frames
6	Exposure (brightness) [19]	The amount of light reaching the surface of the electronic image sensor per unit area
7	Contrast	The disparity in shape, colour, and lighting that exists between several elements of an image
8	Noise [20]	Unpredictable changes in the brightness or colour of frames
9	Slicing [15]	A vision artefact resulting in the loss of selected image slices
10	Flickering [17]	Frequently fluctuating colour or brightness throughout the time dimension

**Table 3 sensors-23-01769-t003:** Results of individual algorithms.

Model Name	Average Value	Largest Value	Smallest Value	Standard Deviation
XGBClassifier **(with tunable parameters)**	**91.60%**	**92.50%**	**90.70%**	**0.005**
XGBClassifier (default)	89.90%	91.10%	88.80%	0.006
AdaBoostClassifier (default)	89.50%	90.80%	88.40%	0.007
SVC (default)	87.50%	89.00%	85.90%	0.007
LogisticRegression (default)	88.20%	89.50%	87.00%	0.006
DecisionTreeClassifier (default)	87.20%	88.20%	86.70%	0.005

**Table 4 sensors-23-01769-t004:** XGBoost results based on sample category.

Sample Category	Precision	Recall	F1-Score
“In the wild” Content	0.8520	0.8655	0.8590
Proffesionally Generated Content	0.9445	0.9370	0.9395

**Table 5 sensors-23-01769-t005:** Percentage deviation of indicators.

	“In the Wild”Correct	“In the Wild”Wrong	PRO Correct	PRO Wrong
Temporal Activity (TA)	8.31%	53.86%	9.67%	133.62%
Flickering	0.24%	1.89%	0.14%	1.57%
Blockloss	2.15%	40.89%	8.92%	161.58%
Blockiness	0.86%	8.56%	0.52%	8.10%
Spatial Activity (SA)	2.00%	9.89%	1.92%	36.75%
Slice	4.21%	103.72%	4.27%	80.48%
Exposure (brightness)	0.43%	0.74%	0.28%	2.01%
Blur	0.43%	3.14%	3.09%	46.71%
Noise	4.74%	43.92%	2.97%	65.56%
Contrast	0.81%	1.01%	0.45%	1.46%

**Table 6 sensors-23-01769-t006:** Results of CNN networks.

Model Name	Accuracy	Precision	Recall	F1-Score
CNN	89.60%	92.16%	93.30%	0.93
CNN-LSTM	89.67%	92.66%	92.78%	0.93

**Table 7 sensors-23-01769-t007:** Classification results of individual algorithms in percentages.

Model Name	Precision	Recall	F1-Score
XGBClassifier **(with tunable parameters)**	**91.75%**	**91.75%**	**0.917**
XGBClassifier (default)	90.00%	90.00%	0.9000
AdaBoostClassifier (default)	89.45%	89.45%	0.8945
SVC (default)	87.30%	87.40%	0.8705
LogisticRegression (default)	87.75%	87.90%	0.8775
DecisionTreeClassifier (default)	87.30%	87.10%	0.8715

## Data Availability

Not applicable.

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
