# Peer review of "“In the Wild” Video Content as a Special Case of User Generated Content and a System for Its Recognition"

_sensors, 2023, doi:10.3390/s23041769_

Round 1

Reviewer 1 Report

The paper proposes a method to classify user generated content vs professional content in the context of videos.

The paper needs major improvements.

Here below, I try to list all needed fixes

1. In the abstract of the paper, the major literature pitfalls or space of improvements or gaps should be emphasized.

2. In the introduction, the authors should make clear statements (or paragraphs) about: i) motivations, ii) objective, iii) overview on the overall approach, (iv) gaps in the literature, (v) which of those gaps has been filled, (vi) contributions to the literature (and to the video analysis research area in general), (vii) novelties of the paper. 

3. In the introduction, I strongly suggest to move the discussion on the definition of Multimedia Quality Assessment to another subsequent section. Moreover, an image can facilitate framing the definition.

4.  It is not clear how authors have collected the samples in the database. Please, discuss deeply the procedure and the technologies used. Please, also add other information on the samples and the database at hand. A table summarizing main database features is recommended.

5. Video indicators must be discussed deeply and motivated. The citations only are not sufficient.

6. Why using only XGBoost? In other papers (e.g., see https://doi.org/10.1371/journal.pone.0250190, https://doi.org/10.1007/s00521-022-07454-4) the authors experiments with a variety of ML and DL methods and compare them. Moreover, in some of those listed, an approach named multi-view learning is recommended. Please, also motivate the choice to use straightforward and traditional methods against those in the papers suggested. Lastly, in the tables 2 and 3 other ML methods are reported. This is quite misleading. Please, correct these parts of the papers.

7. Motivate the validation approach used. See https://doi.org/10.1007/s00521-022-07454-4 for an overview of different validation approaches. 

8. Line 329: "Measurements are made 20 times for each model". This is somehow in touch with the last part of comment 6. In addition, can the authors motivate the choice of 20?

9. Cases of wrongly classified instances must be presented and discussed. 

10. Related work is missing. Please add a proper related work section. Where does the current paper place itself in the current landscape of video analysis and UGC-videos and MQA? Please, also compare the performance of your method against others in the literature. 

11. Limitations must be added and deeply discussed, as well as future works planned.

12. Lines 318-329 must be better presented. Why the subsampling? Which method has been used? What is the dataset from [6]? It must be described. Also, results should be reported in a better way. To what do the authors refer when they state "100% efficiency"? What is the efficiency? How they have measured it? Or it is a typo for "accuracy"?

13. Expected impact of the method should be discussed. Practical implications as well as resources allocation and consumption should be touched.

14. A visual abstract of the proposal should be added.

Author Response

Reviewer 1:

Dear Reviewer. First of all, thank you for taking the time to review our paper. Naturally, we have commented on your comments, and the answers are provided below.

Additionally, we have attached a paper file with the changes highlighted.

    1. In the abstract of the paper, the major literature pitfalls or space of improvements or gaps should be emphasized.

The following sentences have been added: ``Studies on UGC recognition are scarce. According to research in the literature, there are currently no real operational algorithms that distinguish UGC content from other content.''

    2. In the introduction, the authors should make clear statements (or paragraphs) about: i) motivations, ii) objective, iii) overview on the overall approach, (iv) gaps in the literature, (v) which of those gaps has been filled, (vi) contributions to the literature (and to the video analysis research area in general), (vii) novelties of the paper. 

In Section 1, we have introduced the following text:
As contributions of this paper, we:
\begin{itemize}
    \item Present motivation on what the problem is and why is it important (addressed in Section~\ref{sec:intro});
    \item Show our objectives (addressed in Subsection~\ref{sec:scope});
    \item Reveal gaps in the literature (addressed in Subsection~\ref{sec:mqa});
    \item Propose to fill some of these gaps (addressed in Subsection~\ref{sec:def});
    \item Present overview on our overall approach (addressed in Section~\ref{sec:materials});
    \item Show results contributing to the literature and to the video analysis research area in general (addressed in Section~\ref{sec:results});
    \item Discuss novelties of the paper (addressed in Section~\ref{sec:discussion}).
\end{itemize}

    3. In the introduction, I strongly suggest to move the discussion on the definition of Multimedia Quality Assessment to another subsequent section. Moreover, an image can facilitate framing the definition.

The discussion of multimedia quality assessment has been separated into the "Related Work for Multimedia Quality Assessment" subsection. In addition, a figure has been placed there to facilitate framing the definition.

    4.  It is not clear how authors have collected the samples in the database. Please, discuss deeply the procedure and the technologies used. Please, also add other information on the samples and the database at hand. A table summarizing main database features is recommended.

As a summary of the database, we have described the resolutions and their division. We hope this is sufficient.

    5. Video indicators must be discussed deeply and motivated. The citations only are not sufficient.

The ``Video Indicators''' subsection has been rewritten and expanded.

    6. Why using only XGBoost? In other papers (e.g., see https://doi.org/10.1371/journal.pone.0250190, https://doi.org/10.1007/s00521-022-07454-4) the authors experiments with a variety of ML and DL methods and compare them. Moreover, in some of those listed, an approach named multi-view learning is recommended. Please, also motivate the choice to use straightforward and traditional methods against those in the papers suggested. Lastly, in the tables 2 and 3 other ML methods are reported. This is quite misleading. Please, correct these parts of the papers.

Explained content of the tables and clarified the why we compare them. Added explanation why we use XGBoost. 

     7. Motivate the validation approach used. See https://doi.org/10.1007/s00521-022-07454-4 for an overview of different validation approaches. 

Explained validation chosen.

    8. Line 329: "Measurements are made 20 times for each model". This is somehow in touch with the last part of comment 6. In addition, can the authors motivate the choice of 20? 

Explained the choice of 20 and cleared the relation.

    9. Cases of wrongly classified instances must be presented and discussed. 

Added precision/recall data from samples. Added feature importance table and explanation. Added discussion. 

    10. Related work is missing. Please add a proper related work section. Where does the current paper place itself in the current landscape of video analysis and UGC-videos and MQA? Please, also compare the performance of your method against others in the literature. 

Added comaprison to different literature. 

    11. Limitations must be added and deeply discussed, as well as future works planned.

In conclusions (Section ``Discussion'') we have added discussion about the limitations of the model and provided information on possible future directions of work.

    12a. Lines 318-329 must be better presented. Why the subsampling? Which method has been used? What is the dataset from [6]? It must be described.

This paragraph has been rephrased to:
The first test set is a database that originates from \cite{leszczuk2022user}. It uses 3 publicly available ``in the wild`` video databases: (i) CVD-2014~\cite{Nuutinen2016}, (ii) LIVE-Qualcomm~\cite{Ghadiyaram2018}, and (iii) KoNViD-1k~\cite{hosu2017}. The database of video sequences is supplemented with a ``counterweight'' in the form of professional quality video sequences. For this purpose, the ``NTIA simulated news'' database is used~\cite{pinson2013choose}. At the beginning there is a high imbalance of samples in the class (``in-the-wild'' content vs. professional content) of about 1:32, which is not recommended for modelling. Due to this, samples of the second class are randomly subsampled to obtain a more balanced ratio of 1:5. As a result, 408 samples are used for the database, which is divided into 68:304. The model achieves an efficiency of 94.5\% in it.

    12b. Also, results should be reported in a better way. To what do the authors refer when they state "100\% efficiency"? What is the efficiency? How they have measured it? Or it is a typo for "accuracy"?

The results report was updated. The efficiency typo was changed.     

    13. Expected impact of the method should be discussed. Practical implications as well as resources allocation and consumption should be touched.

The description of possible applications and impact has been expanded in the Discussion section. Presentation of the performance of the soltion was moved from section 2.3 to section 3 and expanded.

    14. A visual abstract of the proposal should be added.

A visual abstract of the proposal has been added.

Reviewer 2 Report

In this paper, authors have introduced “In the Wild” Video Content as a Special Case of User Generated Content and a System for Its Recognition. This topic seems to be very interesting and promising. However, I have some queries regarding the proposed technique.

1.       The introduction and the proposed methodology are well described. However, authors are suggested to state the motivation and objective of this paper clearly in the manuscript.

2.       The abstract of this paper needs to be revised in terms of its applications.

3.       Place abbreviations in parentheses following the spelled-out forms the first time they appear in the text. If they are used in the abstract, define them in the abstract and again in the text.

4.       Mathematical symbols—single letters used to designate unknown quantities, constants, and variables are set in italic type and its meaning should be given the first time when they appear in the text.

5.       There are very few recent references available in this paper. Add some more recent references.

6.       The obtained results are very impressive. The number of test suits is quite sufficient. However, some comparisons can improve the quality of this paper.

7.       Author must include the future direction of the research in conclusion section.

8.       Recheck the format of writing equations and presentation

Too many punctuations and grammar in the manuscript. Rectify them

Author Response

Reviewer 2:

Dear Reviewer. First of all, thank you for taking the time to review our paper. Naturally, we have commented on your comments, and the answers are provided below.

Additionally, we have attached a paper file with the changes highlighted.

    1.       The introduction and the proposed methodology are well described. However, authors are suggested to state the motivation and objective of this paper clearly in the manuscript.

In Section 1, we have introduced the following text:
As contributions of this paper, we:
\begin{itemize}
    \item Present motivation on what the problem is and why is it important (addressed in Section~\ref{sec:intro});
    \item Show our objectives (addressed in Subsection~\ref{sec:scope});
    \item Reveal gaps in the literature (addressed in Subsection~\ref{sec:mqa});
    \item Propose to fill some of these gaps (addressed in Subsection~\ref{sec:def});
    \item Present overview on our overall approach (addressed in Section~\ref{sec:materials});
    \item Show results contributing to the literature and to the video analysis research area in general (addressed in Section~\ref{sec:results});
    \item Discuss novelties of the paper (addressed in Section~\ref{sec:discussion}).
\end{itemize}

    2.       The abstract of this paper needs to be revised in terms of its applications.

We added one sentence to the abstract: ``It is provided...''

    3.       Place abbreviations in parentheses following the spelled-out forms the first time they appear in the text. If they are used in the abstract, define them in the abstract and again in the text.

We have expanded the abbreviations pip, CLI and CSV in the manuscript.

    4.       Mathematical symbols—single letters used to designate unknown quantities, constants, and variables are set in italic type and its meaning should be given the first time when they appear in the text.

Currently, there are no single-letter math symbols in the manuscript.

    5.       There are very few recent references available in this paper. Add some more recent references.

Added more recent references. 

     6.       The obtained results are very impressive. The number of test suits is quite sufficient. However, some comparisons can improve the quality of this paper. 

Explained it deeper.

    7.       Author must include the future direction of the research in conclusion section.

In conclusions (Section ``Discussion'') we have added discussion about the limitations of the model and provided information on possible future directions of work.

    8.       Recheck the format of writing equations and presentation

There are currently no equations in the manuscript.

    Too many punctuations and grammar in the manuscript. Rectify them

The manuscript has undergone full proofreading.

Round 2

Reviewer 1 Report

I appreciate the effort of the authors in replying to all my comments. Good job!

- I suggest authors to perform proofreading of their paper for finding some errors/typos like survay instead of survey and so on.

- Moereover, I suggest authors to put examples of UGC and PRO with regards to the features listed in table 2 (perhaps in the appendix) and add inside this table another column where to describe the feature.

- Figure 3 is too big with respect to the font size of the text as well as Figure 4. Please adjust those texts. Furthermore, add more descriptive captions to those figures.

- Please, move the paragraph about the python pakage in the appendix of the paper.

Author Response

Reviewer 1:

Dear Reviewer. First of all, thank you for taking the time to review our paper. Naturally, we have commented on your comments, and the answers are provided below.

Additionally, we have attached a paper file with the changes highlighted.

    - I suggest authors to perform proofreading of their paper for finding some errors/typos like survay instead of survey and so on.

The entire manuscript was reviewed.

    - Moereover, I suggest authors to put examples of UGC and PRO with regards to the features listed in table 2 (perhaps in the appendix) and add inside this table another column where to describe the feature.

Frames from two sample shots are attached along with their indicators. Indicators are described.

    - Figure 3 is too big with respect to the font size of the text as well as Figure 4. Please adjust those texts. Furthermore, add more descriptive captions to those figures.

Fixed figures (enlarged font).

    - Please, move the paragraph about the python pakage in the appendix of the paper.

Paragraph about the Python package moved to the Appendix.

Reviewer 2 Report

Authors have incorporated all the changes , Now it is ready for publications. 

Author Response

Reviewer 2:

Dear Reviewer. Thank you for taking the time to review our paper.